# Human Breast Milk microRNAs, Potential Players in the Regulation of Nervous System

**DOI:** 10.3390/nu15143284

**Published:** 2023-07-24

**Authors:** Luis Freiría-Martínez, Marta Iglesias-Martínez-Almeida, Cynthia Rodríguez-Jamardo, Tania Rivera-Baltanás, María Comís-Tuche, Daniela Rodrígues-Amorím, Patricia Fernández-Palleiro, María Blanco-Formoso, Yolanda Diz-Chaves, Natalia González-Freiria, María Suárez-Albo, Montserrat Martín-Forero-Maestre, Cristina Durán Fernández-Feijoo, Jose Ramón Fernández-Lorenzo, Ana Concheiro Guisán, Jose Manuel Olivares, Carlos Spuch

**Affiliations:** 1Translational Neuroscience Group, Galicia Sur Health Research Institute (IIS Galicia Sur), SERGAS-UVIGO (Servizo Galego de Saúde-Universidade de Vigo), 36312 Vigo, Spain; luis.freiria@iisgaliciasur.es (L.F.-M.); marta.iglesias@iisgaliciasur.es (M.I.-M.-A.); cynthia.rodriguez@iisgaliciasur.es (C.R.-J.); tania.rivera@iisgaliciasur.es (T.R.-B.); maria.comis@iisgaliciasur.es (M.C.-T.); damorim@mit.edu (D.R.-A.); patricia.palleiro@iisgaliciasur.es (P.F.-P.); jose.manuel.olivares.diez@sergas.es (J.M.O.); 2Department of Functional Biology and Health Sciences, Campus Lagoas Marcosende, Universidade de Vigo, 36310 Vigo, Spain; 3Picower Institute for Learning and Memory, Massachusetts Institute of Technology, Cambridge, MA 02139, USA; 4Department of Physical Chemistry, Singular Center for Biomedical Research (CINBIO), Universidade de Vigo, 36310 Vigo, Spain; maiblancoformoso@uvigo.es; 5Laboratory of Endocrinology, Singular Center for Biomedical Research (CINBIO), Universidade de Vigo, 36310 Vigo, Spain; yolandadiz@uvigo.es; 6Neonatal Intensive Care Unit, Alvaro Cunqueiro Hospital, 36312 Vigo, Spain; natalia.gonzalez.freiria@sergas.es (N.G.-F.); maria.suarez.albo@sergas.es (M.S.-A.); jose.ramon.fernandez.lorenzo@sergas.es (J.R.F.-L.); ana.concheiro.guisan@sergas.es (A.C.G.); 7Human Milk Bank of Vigo, Alvaro Cunqueiro Hospital, 36312 Vigo, Spain; montserrat.martin.-.forero.maestre@sergas.es; 8Neonatology Unit, University Hospital of Santiago de Compostela (CHUS), 15706 Santiago de Compostela, Spain; cristina.duran.fernandez.feijoo@sergas.es; 9CIBERSAM (Network Biomedical Research Center on Mental Health), 28029 Madrid, Spain

**Keywords:** exosome, human milk, colostrum, miRNAs, small RNA sequencing, nervous system, neurodevelopment

## Abstract

Human milk is the biological fluid with the highest exosome amount and is rich in microRNAs (miRNAs). These are key regulators of gene expression networks in both normal physiologic and disease contexts, miRNAs can influence many biological processes and have also shown promise as biomarkers for disease. One of the key aspects in the regeneration of the nervous system is that there are practically no molecules that can be used as potential drugs. In the first weeks of lactation, we know that human breast milk must contain the mechanisms to transmit molecular and biological information for brain development. For this reason, our objective is to identify new modulators of the nervous system that can be used to investigate neurodevelopmental functions based on miRNAs. To do this, we collected human breast milk samples according to the time of delivery and milk states: mature milk and colostrum at term; moderate and very preterm mature milk and colostrum; and late preterm mature milk. We extracted exosomes and miRNAs and realized the miRNA functional assays and target prediction. Our results demonstrate that miRNAs are abundant in human milk and likely play significant roles in neurodevelopment and normal function. We found 132 different miRNAs were identified across all samples. Sixty-nine miRNAs had significant differential expression after paired group comparison. These miRNAs are implicated in gene regulation of dopaminergic/glutamatergic synapses and neurotransmitter secretion and are related to the biological process that regulates neuron projection morphogenesis and synaptic vesicle transport. We observed differences according to the delivery time and with less clarity according to the milk type. Our data demonstrate that miRNAs are abundant in human milk and likely play significant roles in neurodevelopment and normal function.

## 1. Introduction

Human breast milk (HBM) is a gold standard for preterm and term infant nutrition. It combines all essential nutrients and bioactive factors [1], either synthesized in the lactating breast or transferred via systemic circulation. It provides essential functions for the lactating mother and the breastfed infant [2]. HBM composition is dynamic and may vary according to various factors related to mother, infant, environment and physiology [1]. Among the whole variety of bioactive components, it includes miRNAs [2].

miRNAs are small, 19–24 nucleotide (nt), highly conserved, noncoding RNAs that play critical and essential roles by modulating gene expression through their interaction with cellular mRNAs [3,4]. miRNAs have a high presence in many human body fluids [4]. HBM appears to be one of the richest sources of microRNA, which is also highly conserved in its different fractions [2,5]. Human breast tissue has a specific miRNA expression profile [6], and HBM miRNAs arise primarily from the mammary gland epithelium rather than maternal circulation [7]. Then, they are thought to enter the systemic circulation of the fed infant and exert tissue-specific epigenetic regulation of various functions, including immunoprotecting and development [2,8].

As key regulators of gene expression networks in normal physiologic and disease contexts, miRNAs can influence many biological processes and have also shown promise as biomarkers for disease [9]. miRNAs fulfil various developmental functions, including forming specialized organelles in neuronal and other differentiated cells [10]. Many of them are dynamically regulated during the maturation of the central nervous system (CNS), and they are spatially expressed in the adult brain, indicating essential roles in neural expansion and function, and highlighting the versatile roles of miRNAs in normal CNS physiology and their association with several types of neurological diseases [11,12,13,14]. miRNAs may also be active in the adult brain to coordinate gene function associated with learning and memory [15].

Previous research in milk miRNA has focused not only on HBM [3,16,17,18] but also on animal milk, including bovine [19,20], porcine [21,22,23], or goat [24]. miRNAs are present as free molecules in whole milk [18,25], in all milk fractions [26], skim milk [27], cells [28], and fat [3], but also packaged in vesicles such as exosomes [17]. 

Exosomes are small cell-derived vesicles present in body fluids and carrying proteins and molecules. They have been shown to take up miRNAs mediating their protection against digestion and facilitating their regulatory functions in different tissues and organs [19,26,29]. Exosomes have been shown to possess specific types and levels of miRNAs compared to those found overall in cells under various conditions. This supports the hypothesis that the miRNA packaging process is a selective rather than a random event [8]. Exosomes may be essential for the long-distance transport of microRNA, given that they are surrounded by a lipid bi-layered membrane and are equipped with adherence molecules, both of which facilitate their ordered endosomal transfer via epithelial cells of the intestine [2,30]. 

When exposed to acidic conditions that mimic gastric and pancreatic digestion, milk exosomes prevent the degradation of vulnerable miRNAs [31]. Studies have reported that milk exosomes protect exosomal miRNAs from RNase enzymatic, chemical, or mechanical degradation [20,31,32]. Subsequently, milk exosomes are taken up by intestinal epithelial cells by endocytosis and moved into the systemic circulation [17,18]. microRNAs are transferred among animal species by dietary means, compelling evidence that the intestinal uptake of microRNAs encapsulated in exosomes is an active, saturable process most likely mediated by endocytosis [30]. 

Given the physiological activity, safety, biocompatibility, and drug delivery potential, the application of milk exosomes in therapeutics is innumerable [32], being a novel therapeutic option to prevent oxidative stress [33], inflammation [15], ulcerative colitis [34] or necrotizing enterocolitis [35,36]. Moreover, milk-derived miRNAs can enter normal and cancer cells and change their miRNA expression profile [24]. 

miRNAs in HBM exosomes may help modulate several biological processes, such as neurodevelopment. In the present study, we aimed to investigate, through miRNA-sequencing analysis, the miRNA content of exosomes in HBM and how they contribute to CNS development.

miRNAs are small and well-conserved molecules capable of regulating posttranscriptional gene expression in the central nervous system. The expression dynamics of these miRNAs in neurodevelopment correctly activate genes that regulate many biological pathways in neurons and glia. Importantly, these miRNAs are spatially expressed in the adult brain, indicating their essential roles in neuronal development and function. This makes us consider what combinations of miRNAs could be regarded for the recovery of neurological damage or the regulation of neural networks in neuropsychiatric pathologies. That is why we propose this work, where we investigate a very rich source of miRNAs, such as human milk, potential miRNAs that are capable of regulating pathways in the nervous system.

## 2. Materials and Methods

### 2.1. Milk Sample Collection

Human breast colostrum (0–5 days postpartum) and mature milk (1–6 months postpartum) from term (T) mothers (≥37 weeks of gestation), mothers of late preterm (LPT) infants (34–36 weeks of gestation), and mothers of moderate/very preterm (MVPT) infants (28–34 weeks of gestation) were collected from the Human Milk Bank of Vigo, located in Álvaro Cunqueiro Hospital of Vigo, Spain. Samples were stored at −20 °C until exosome extraction. This project was approved by Galician Ethics Committee (Register code 2016/559). 

Thirty-six samples were collected, and five groups were made according to the time of delivery and milk states: mature milk (Group 1; *n* = 8) and colostrum at term (Group 2; *n* = 10); moderate and very preterm mature milk (Group 3; *n* = 8), moderate and very preterm colostrum (Group 4; *n* = 6) and late preterm mature milk (Group 5; *n* = 4). 

### 2.2. Exosome Extraction

Exosome extractions were done with EX04 Exo-spin^TM^ midi column (Cell Guidance Systems, Cambridge, UK), optimized by our laboratory for human BM. 1 mL of each milk sample was centrifuged at 16,000× *g* for 30 min at 4 °C to remove cells and cellular debris. The entire supernatant, including the milk fat layer, was transferred to a new 2 mL centrifuge, and Exo-spin^TM^ Buffer was added in a 1:1 ratio. After mixing by vortex, the tube was incubated at 4 °C overnight using a tube rotator. The mixture was centrifuged at 16,000× *g* for 60 min at 4 °C. Carefully. The supernatant was aspirated and discarded. The exosome-containing pellet was resuspended in 1 mL of PBS to avoid sample drought. 

After Exo-spin^TM^ midi column preparation and equilibration, 500 µL of the whole exosome-containing sample was applied to the top of the column. The fraction was collected in a 1.5 mL microcentrifuge tube by gravity and labelled as Fraction 1. Rest 500 µL of the sample was applied to the top of the column and collected by gravity in a new 1.5 mL microcentrifuge tube (Fraction 2). The process was repeated, loading 500 µL of PBS 21 times until 24 of 500 µL each were collected. Fractions 1–24 were stored at −20 °C until further steps.

Exosome-containing fractions from each sample were diluted 1:1 in Sample Buffer Laemmli Concentrate (S3401; Sigma-Aldrich, San Luis, MO, USA) for denaturing and boiled at 95 °C for 5 min. 20 µL of diluted samples were loaded in 8–14% Bis-Tris polyacrylamide gels, followed by electrophoretic transfer of proteins. The membranes were then incubated overnight at 4 °C with primary antibodies for markers of exosomes: anti-CD9 (C-4) mouse monoclonal antibody 1:200 (sc-13118; Santa Cruz Biotechnology, Dallas, TX, USA), anti-CD63 (CLB-180) mouse monoclonal antibody 1:200 (sc-59284; Santa Cruz Biotechnology, Dallas, TX, USA) and anti-CD81 mouse monoclonal antibody 1:200 (sc-70803; Santa Cruz Biotechnology, Dallas, TX, USA) (Figure 1).

### 2.3. Transmission Electron Microscopy (TEM)

Exosomes were visualized by negative staining following methods from Jung et al. 2018 [37]. Purified exosomes were previously diluted 1/20 to avoid high concentration and fixed with 1 mL of 2% Paraformaldehyde (PFA) for 5 min. 5 µL of exosome suspension was loaded on a 300-mesh carbon/formvar-coated grid and dried at room temperature. Next, the grids were washed with distilled sterile water and 20 drops of filtered 1% uranyl acetate (UA) solution on the surface of the electron microscopy grid for 10 min. After staining, the grid was rinsed with water to remove the excess staining solution. The grid was placed on the table by holding with tweezers and covered the grid partially with a culture dish to dry for 10 min. Samples were imaged using a JEOL JEM 1010 High Contrast Transmission Electron Microscope (JEOL, Akishima, Tokyo, Japan), operated at 100 kV (Figure 2).

### 2.4. RNA Extraction and Quality Control

Total RNA was extracted and purified from one selected exosome-containing fraction from each processed milk sample using a miRNeasy Mini Kit (Qiagen, Hilden, Germany) following the manufacturer’s protocol. BioDrop µLite 7141 V1.0.3 (BioChrom, Cambridge, UK) was used to determine 260/280 and 260/230 purity ratios. The quantity and integrity of the RNA were assessed using an RNA 6000 Nano Kit (Agilent, Santa Clara, CA, USA), and the % of the miRNA fraction was measured using and Small RNA Kit on a 2100 Bioanalyzer (Agilent Technologies, Santa Clara, CA, USA).

### 2.5. microRNA-Sequencing

Libraries were prepared by RealSeq Biosciences (Santa Cruz, CA, USA) using 100 ng of RNA input (based on the concentrations measured by the customer) and 20 cycles of PCR following the manufacturer’s recommendations. The libraries were pooled to equal nanomolarity concentrations and then purified, and the size was selected using Pippin Prep (Sage Biosciences, Beverly, MA, USA). The library pool was profiled using a Tapestation and Qubit before sequencing on the NextSeq 550 v2 High-Output—SR 75 Cycle (Illumina, San Diego, CA, USA).

The raw fastq files were initially processed using Cutadapt 2.3 with Python 3.6.8 (TU Dortmund University, Dortmund, Germany). Adapter sequences were removed, and reads were filtered based on length. Reads shorter than 5 bp were filtered first to determine RNA degradation. Reads with a minimum 15 bp length were aligned to the human dataset in miRBase with Bowtie [38]. Samtools 1.7, using htslib 1.7–2, was used to convert SAM files to BAM. The ‘BuildBamIndex’ picard-tool via the PicardCommandLine was used to generate an index and convert BAMs to reads files. A count matrix was made using a proprietary Python program [39]. After obtaining a counts matrix of sample vs. aligned-object clustering, differential expression analysis was performed using the DESeq2 software, which models the data using a negative binomial distribution [40]. Plots were generated using the ggplot two package in R [41].

### 2.6. miRNA Functional Assays and Target Prediction

Functional Gene Ontology (GO) enrichment and Kyoto Encyclopaedia of Genes and Genomes (KEGG) pathway analysis were performed by using miRPathDB 2.0 (https://mpd.bioinf.uni-sb.de/overview.html accessed on 17 March 2021). GO, and KEGG categories were selected when *p*-value < 1 × 10^−3^. miRNA targets were studied after selecting enriched GO and KEGG categories related to the nervous system and analysing gene targets implicated in that neuro-related terms.

## 3. Results

One of the innovations of this work is that we use milk samples taken under different conditions of the mother. We work with milk from mothers who delivered at term (≥37 weeks of gestation), mothers who delivered at late preterm pregnancy (34–36 weeks of gestation) and mothers who had moderate or very preterm deliveries (28–34 weeks of gestation). Another important aspect of this work is that we include two other types of breast milk. On the one hand, colostrum, the first milk produced by the breast, begins to be generated in the middle of pregnancy (12–18 weeks) and continues during the first days after the baby’s birth, and on the other hand, mature HBM.

In this way, we seek to describe the types of miRNAs that appear in these types of milk and, on the other hand, increase the chances of finding miRNAs that regulate the nervous system. This period is when important genetic and epigenetic changes occur in the development of the unborn child’s brain.

### 3.1. RNA Extraction and Quality Control

We previously isolated exosomes from HBM and confirmed its presence by using transmission electron microscopy (Figure 2) and Western blot for exosome-specific surface markers (Figure 1). We extracted total RNA from the exosome-containing fractions and examined the RNA concentration, 260/280 and 260/230 purity ratios. Next, we measured the RNA integrity number (RIN) and percentage of small RNA fraction (<200 nt) that miRNAs (19–25 nt) represent to gain a more detailed view of RNA integrity within exosomes (Table 1). 

The average RNA concentration extracted from exosome-containing samples was 17.63 ± 7.92 ng/µL, ranging in most samples between 20 and 10 ng/µL and without clear differences across the groups. The ribosomal RNA (rRNA) signal was very low or almost absent (Figure 3), so RINs were, in most cases, below 2.

Small RNA fraction was present, and when we measured the % of miRNA fraction, that generally was more than 80% (Figure 4).

### 3.2. miRNA Differential Expression Analysis

The goal of differential expression analysis is to determine whether differences in the expression of each miRNA between groups are significant, given the amount of variation observed within groups. RNA sequencing identified a total of 132 different miRNAs within all groups analysed. Due to the differences in the sample types, we performed six paired comparisons (Table 2).

Comparison between term mature milk (G1) and term colostrum (G2) (Appendix A) 118 different miRNAs were identified, from which 11 miRNAs had significant (*p*-value < 0.05) different expressions. 11 miRNAs had significant (*p*-value < 0.05) different expression between G2 and G1. 7 miRNAs were upregulated (miR-4516, miR-200c-3p, miR-205-5p, miR-141-3p, miR-486-p, miR-3960 and miR-4492) in term colostrum. 4 (miR-151b, miR-151a-5p, miR-6731-3p and miR-29a-3p) had lower expression in the same group, and thus big expression in term mature milk (Figure 5).

Moderate/very preterm mature milk (G3) vs. term mature milk (G1) comparison (Appendix A) showed 99 different identified miRNAs. 24 of them were significantly different expressed (*p*-value < 0.05) in G3 vs. reference group G1. 12 miRNAs had big expression in moderate/very preterm mature milk than in term mature milk: miR-3960, miR-1246, miR-10400-5p, miR-1249-3p, miR-7874-3p, miR-7704, miR-6731-3p, miR-1290, miR-23b-3p, miR-23a-3p, miR-3196, miR-1908-3p and miR-6836-3p. Other 12 miRNAs (miR-17-5p, miR-16-5p, miR-486-5p, miR-93-5p, let-7g-5p, let-7c-5p, let-7a-5p, miR-20a-5p, miR-146b-5p, let-7d-5p, miR-146a-5p and miR-451a) were downregulated in G3 vs. G1 (Figure 6).

For different expression analyses between term mature milk (G1) vs. late preterm mature milk (G5) (Appendix A), 105 different miRNAs were identified. In the G5 comparison vs. reference group G1, 15 miRNAs had significantly different expression (*p*-value < 0.05). All of the 15 miRNAs (miR-6126, miR-7847-3p, miR-6731-3p, miR-877-3p, miR-451a, miR-486-5p, miR-4488, miR-320c, miR-144-3p, miR-3196, miR-1249-3p, miR-6090, miR-93-5p, miR-423-5p, miR-3184-3p) significantly different expressed in late preterm mature milk group in comparison with term mature milk were downregulated (Figure 7).

Different expression analysis between moderate/very preterm colostrum (G4) and term colostrum (G2) (Appendix A) 137 different identified miRNAs were identified. By comparing G4 with G2 as the reference group, 27 miRNAs had significantly different expression (*p*-value < 0.05). 15 miRNAs had big expression in the term colostrum group than in moderate/very preterm colostrum: miR-3178, miR-29a-3p, miR-29b-3p, miR-342-3p, miR-29c-3p, miR-99b-5p, miR-4510, miR-148a-3p, miR-25-3p, miR-125b-5p, miR-106b-5p, miR-151a-5p, miR-1290, miR-30c-5p and miR-6731-3p. 12 miRNAs are downregulated within the same comparison: miR-144-3p, miR-451a, miR-671-5p, miR-12136, miR-30b-5p, miR-320a-3p, miR-320b, miR-378a-3p, let-7b-5p, miR-378c, miR-486-5p and miR-320c (Figure 8).

Comparing moderate/very preterm colostrum (G4) vs. moderate/very preterm mature milk (G3) (Appendix A), 131 different miRNAs were identified. 19 of them had significant (*p*-value < 0.05) different expression in G4 with G3 as reference group. Fifteen miRNAs were upregulated in moderate/very preterm colostrum group (miR-200c-3p, miR-16-5p, miR-3178, miR-20a-5p, miR-17-5p, miR-25-3p, let-7d-5p, let-7c-5p, let-7a-5p, miR-195-5p, miR-26a-5p, miR-103a-5p, miR-103b, miR-107 and miR-30a-5p). Only four miRNAs (miR-10400-5p, miR-375-3p, miR-3960 and miR-203b-5p) had lower expression in moderate/very preterm colostrum than in moderate/very preterm mature milk (Figure 9).

Different expression analysis between late preterm mature milk (G5) and moderate/very preterm mature milk (G3) (Appendix A) rose 88 different miRNAs. By comparing G5 vs. reference group G3, 36 of them were significantly different expressed (*p*-value < 0.05). 11 miRNAs (miR-26a-5p, miR-378c, miR-378a-3p, let-7g-5p, let-7a-5p, let-7c-5p, let-7d-5p, miR-181b-5p, let-7i-5p, miR-146a-5p and miR-146b-5p) had higher expression in late preterm than in moderate/very preterm mature milk. Twenty-five miRNAs (hsa-miR-7847-3p, hsa-miR-6126, hsa-miR-6090, hsa-miR-3196, hsa-miR-877-3p, hsa-miR-1249-3p, hsa-miR-1246, hsa-miR-3960, hsa-miR-6731-3p, hsa-miR-1290, hsa-miR-1908-3p, hsa-miR-375-3p, hsa-miR-5787, miR-6836-3p, miR-7704, miR-4488, miR-423-5p, hsa-miR-3184-3p, hsa-miR-4516, hsa-miR-181a-5p, hsa-miR-203a-3p, hsa-miR-29c-3p, hsa-miR-203b-5p, hsa-miR-10400-5p and hsa-miR-4634) were downregulated in late preterm mature milk in comparison with moderate/very preterm mature milk (Figure 10).

In summary, after the whole set of comparisons, 132 different miRNAs were identified after RNA sequencing. Sixty-nine (52.27%) had significantly different expressions, at least in one comparison analysis. Higher numbers of differentially expressed miRNAs were found when comparing groups according to the delivery time (preterm versus term) compared to milk type (colostrum versus mature milk). Differential expressed analysis having late preterm mature milk (G5) gave the 2nd lower number of miRNAs (15 when compared to term mature milk), but also the comparison with the highest number of miRNAs (36 after comparison with moderate/very preterm mature milk) (Table 2). 

### 3.3. Principal Component Analysis

Principal Component Analysis (PCA) emphasises variation into and between groups. More concretely, it allows us to determine groups’ variation and how the samples from each group cluster together. Across the six paired comparisons, we saw big variability in and between almost all milk types. Milk samples within term mature milk and moderate/very preterm mature milk and colostrum groups had distinctly different miRNA profiles. No clear clustering was seen for samples belonging to them. However, clustering can be seen for late preterm mature milk (G5) samples (Figure 11).

### 3.4. miRNA GO and KEGG Pathway Analysis

To better understand the role of miRNAs in HBM exosomes, potential targets of miRNAs at the human genome level were explored by using the online tool miRPathBD (https://mpd.bioinf.uni-sb.de/overview.html accessed on 24 July 2021). We first selected the significative categories with predicted or experimental target evidence with at least a *p*-value < 10^−3^. After focusing on brain and nervous system terms, redundant terms were deleted and summarized.

Within the term colostrum and mature milk comparison, milk type slightly differs. Upregulated miRNAs seem to have a little bit more impact over GO and KEGG categories than those that are downregulated (Figure 12). 

For cellular components, all miRNAs were enriched in the “Synapse” term. miR-4492 has targets in all categories except for “Dendrite”. The biological process “Nervous system development” is the most regulated by differential expressed miRNAs, but also “Neurogenesis”, “Neuron projection morphogenesis” and “Chemical synaptic transmission” appear as targets. KEEG pathways terms were almost only assigned to miR-29a-3p.

Preterm delivery differentially augments the expression of miRNAs that control neuronal components and neuron formation (Figure 13). Upregulated miRNAs in moderate/very preterm compared to term mature milk have a higher number of annotations for neuron-located cellular components and neurodevelopmental biological processes than downregulated miRNAs. “Synapse” is the most enriched GO cellular component. The rest of the terms have many miRNAs associated but only in the upregulated set. Biological processes like “Nervous system development”, “Neurogenesis” and “Neuron projection morphogenesis” are the more abundant terms. miR-3196 has the wider range of annotations, but also miR-1249-3p, miR-7847-3p and miR-1908-3p have an important variety of targets. This set of downregulated miRNAs has more matches in terms of KEGG pathways and more concretely for “Glioma” or “Spinal cord injury”.

All of the differentially expressed miRNAs between late preterm and term mature milk were downregulated, and this set has a substantial number of targets implicated in components of neurons and processes that contribute to nervous system formation and maturation (Figure 14). 

“Synapse” is the more abundant cellular component, but the rest of the terms are also associated with many miRNAs. For biological processes, “Neurogenesis”, “Neuron projection morphogenesis” and “Nervous system development” are the most enriched terms, but also “Synaptic vesicle cycle”, “Neurotransmitter transport” and “Glioma” are targets of several miRNAs. miR-7847-3p, miR-4488, miR-3196, miR-423-5p, miR-1249-3p and miR-6090 are the miRNAs that have more enriched terms across all categories.

Colostrum comparative between moderate/very preterm and term miRNA hits shows quite a balance between the effect of upregulated and downregulated miRNAs over cellular components and biological processes implicated in the nervous system. “Synapse” is the most enriched term, and “Axon” is abundant across all differentially expressed miRNAs. miR-4510 has targets in all cellular component categories, and miR-148a-3p only failed for the “Neuronal cell body” term. For biological processes, “Neurogenesis”, “Neuron projection morphogenesis” and “Nervous system development” are the most enriched terms, pointing to miR-4510 and miR-671-5p as the ones with more matched terms.

The upregulated set of miRNAs has many regulated KEGG pathways, especially those controlled by miR-29a-3p and miR29b-3p (Figure 15).

Milk type has a clearer effect when compared to moderate/very preterm milk types (colostrum vs. mature milk) than term comparison. Upregulated miRNAs in moderate/very preterm colostrum have more impact on neuronal compartments and development, even though downregulated miRNAs also target several GO terms. For cellular components, “Synapse” is the more abundant term, but “Neuronal projection”, “Axon”, “Dendrite” and “Somatodendritic compartment” also match for several miRNAs. miR-16-5p, miR-103a-5p, miR-107 and miR-195-5p have more enriched terms. “Nervous system development” is the category that has more miRNAs targets. In contrast, KEGG pathways are only regulated by miRNAs more expressed in colostrum than in mature milk, with miR-16-5p and miR-195-5p as the most enriched miRNAs (Figure 16).

Downregulated miRNAs on late preterm vs. moderate/very preterm mature milk have shown many hits for neural cellular components and biological processes (Figure 17). The results show that form all these 69 miRNAs, only 3 miRNAs (let-7c-5p, let-7d-5p and miR-99b-5p) had no match for terms related to the nervous system. Thus, 66 miRNAs had at least one predicted target for brain processes.

GO and KEGG annotation analysis suggested that most human milk exosome genes might function in nervous system maturation and establishment of neuronal networks. This pattern is consistent between all the comparatives performed. For the Cellular Component, most miRNAs had targets located in “Synapses”, no matter how they were regulated, and which kind of comparison was performed. Related to that term, “Axon”, “Dendrite” or “Neuron projection” are also frequent annotations. Biological Processes like “Neurogenesis”, “Neuron projection morphogenesis”, and “Nervous system development” are the most related to differentially expressed miRNAs, but also “Chemical synaptic transmission” and “Synaptic vesicle cycle” are regulated processes too. KEGG pathway analysis showed that the “Neurotrophin signaling pathway”, “Glioma”, and “Axon guidance” are the main terms having targets for differentially expressed miRNAs.

In summary, differences between colostrum and mature milk samples were clearer in moderate/very preterm comparisons (G4 vs. G3) than in term comparisons (G2 vs. G1), where differences were less defined. In addition, differences between delivery times appear with lower intensity in colostrum (G4 vs. G2) than in mature milk samples (G3 vs. G1). Thus, miRNA changes and their impact on the nervous system are clearer in moderate/preterm mature milk.

The late preterm mature milk group (G5) is different from both mature milk groups (G1 and G3), having a whole or a majority of miRNAs downregulated that, in addition, are implicated in GO and KEGG annotations related to the nervous system.

### 3.5. miRNA Targeting Neuro-Related Protein Selection

GO and KEGG pathway analysis have shown that many identified miRNAs were predicted to regulate neurodevelopment processes by targeting genes that codify proteins in almost all neuron parts (especially in synapses)—moreover, several miRNAs target brain-related physiological and pathological pathways. To further understand these roles, we perform a new miRNA target analysis for genes codifying nine neuro-related proteins previously identified and quantified by our lab as exosomes cargo (*NRP1*, *CALR*, *CLU*, *SERPINF1*, *APOE*, *KLK6*, *RHOA*, *DPSYL2*, *ANXA5*).

We analyzed target genes like Neuropilin-1 (*NRP1*) and Ras homolog family member A (RHOA), which play roles in axon elongation inhibition and dendrite branching promotion, and Dihydropyrimidinase-related protein 2 (*DPYSL2*), which acts in a contrary way. Serpin family f member 1 (*SERPINF1*) is the codifying gene for the neurotrophic pigment-epithelium derived factor (PEDF). *APOE* encodes Apolipoprotein E, the main cholesterol carrier in the brain, which also affects neuronal growth, membrane repair, and axon formation. ApoE is also implicated in Aβ clearance and is considered an Alzheimer’s disease (AD) risk factor. Calreticulin (CALR) and Clusterin (CLU) codify for chaperones that act against Aβ. CALR is also a Ca^2+^ storage. *KLK6* is the gene of Kallikrein-6/Neurosin, a protease that degrades α-Syn and contributes to neuronal reparation and nerve regeneration. Finally, ANXA5 (Annexin A5) is implicated in Parkinson’s disease (PD) and AD by controlling intracellular Ca^2+^ levels, attenuating α-Syn nuclear translocation and reducing Aβ aggregation.

In colostrum and mature milk comparison, almost all the differentially expressed miRNAs target the chaperone *CLU* gene, sharing some regulators with *CALR*. *NRP1*, *RHOA* and *DPYSL2* have in this panel similar miRNAs. miR-141-3p is the one that has more targets. No miRNAs targeting *SERPINF1* and *ANXA5* and no gene targets for miR-486-5p and miR-3960 (Figure 18).

For the set of miRNAs compared between moderate/very preterm and term mature milk (Figure 19), *NRP1* and *CLU* have more targets in upregulated miRNAs, whereas *CLU* again, *CALR*, *RHOA* and *DPYSL2* are the genes with more targets in downregulated miRNAs that, in addition, are the same.

Comparing late preterm and term mature milk show that the *CLU* gene is targeted by the majority of the miRNAs (Figure 20). Also, *NRP1* and *DPYSL2* are targets for several miRNAs. *CALR* and *APOE* genes also have some regulating miRNAs. No miRNA regulating *ANXA5*. No targets for miR-451a, miR-486-5p and miR-1249-3p.

Regulation of *CLU* and *DPYSL2* genes is consistent across up and downregulated miRNAs in colostrum comparison. In addition, *NRP1* has a strong number of miRNAs targeting it, but especially focused on those with lower expression in moderate/very preterm vs. term colostrum. *NRP1* and *DPYSL2*, and the *RHOA* gene, although it is less regulated, share many of the targeting miRNAs. Again, no targets for *ANXA5* (Figure 21). miR-12136 is the one that has more targets in the set of studied genes.

Analyses of moderate/very preterm groups show that *CLU*, *DPYSL2* and *NRP1* are the most enriched in targets. *RHOA* miRNA targets are shared with them. Neither *APOE* nor *ANXA5* are targets for these miRNAs. miR-16-5p and miR-203b-5p are the miRNAs which regulate more genes (Figure 22).

For differentially expressed miRNAs between late preterm and moderate/very preterm mature milk, *NRP1*, *CLU,* and *DPYSL2* are the most regulated targets. Also, *CALR* and *APOE* genes are under the control of several miRNAs. miR-5787 and miR-7847-3p are the ones that regulate more analysed genes (Figure 23).

The results suggest that HBM miRNAs regulate humans’ brain networks and neurodevelopment. In addition, several miRNAs shared the same target genes, proposing to play a key role in brain function and development.

## 4. Discussion

Development of the mammalian nervous system is the result of a series of coordinated events, including transcriptional networks, cell signaling, cell actions and structural organization [13]. These highly orchestrated processes of neural development start with the proliferation of glia and neurons and their migration, followed by programmed cell death, formation of synapses, myelination, and establishment of neuronal circuits [42]. Growth factors play an integral role in neuronal innervation. This process reaches its highest level during the prenatal period, as neurons first establish synaptic contacts [43]. Synapse formation, stabilization, and plasticity are key features of neuronal development [13]. These dramatic morphological rearrangements, which also happen during adolescence, occur with the hypothesized goal of increasing the efficiency of synaptic transmission [43]. Among the sophisticated multi-layered regulatory networks in the nervous system, miRNAs emerge as crucial regulators during embryonic development, adult neurogenesis and upon environmental stimuli [13]. Differentially expressed and broad miRNA patterns across both temporal and spatial dimensions and between male and female prefrontal cortex suggest critical roles for the identified miRNAs in transcriptional networks of the developing human brain and neurodevelopmental disorders [44].

Synapses are increasingly believed to be essential players in the etiology of several neurological diseases [13]. Enrichment analysis using miRPathBD showed the GO term “Synapse” as a target for most differentially expressed miRNAs, as described in the bibliography, too [3]. We can understand this as a very tight regulation of the synapse formation process. Synaptic plasticity is the ability to change synaptic strength. It can result from specific patterns of activity and the alteration of the number of neurotransmitter receptors [13], and many miRNAs are involved in synapse plasticity and their implications for cognitive function and neuronal disorders [45]. We surveyed milk-derived miRNAs that have an impact on neurodevelopment processes by targeting not only synapse components but also components implicated in axon guidance and extension as *DPYSL2* or *NRP1* [46,47] or ApoE protein, which is the main cholesterol carrier in the brain, which also affects to neuronal growth, membrane repair, axon formation [48]. The presence of these miRNAs in the HBM proposes that early brain development may benefit from these functions, and it grows the importance of breastfeeding as a contributor to newborn health.

HBM miRNAs originate from mammary gland epithelium [7]. They are suggested to be transferred to the infant’s bloodstream, and it is further facilitated by the known packaging of milk miRNAs in “vehicle” structures, such as exosomes and other microvesicles, which may be essential for the long-distance transport and for surviving proteolytic digestion [30,31]. In this work, we corroborate that HBM exosomes contain RNA and small RNA (<200 nt) in all groups of breast milk. Large amounts of small RNA suggested that exosomes may contain miRNAs (19–24 nt) [16,18,24,25,27,39]. In addition, we found that rRNAs 18S and 28S were mostly absent or low, which is also consistent with previous publications [2,3,19,20,21,23,29,49].

Being surrounded by a lipid bi-layered membrane and equipped with adherence molecules facilitates ordered endosomal transfer via epithelial intestine cells [30]. RNA presence inside the exosomes can also be considered a protective mechanism against the relatively higher RNase activity reported in milk [19]. Through these vesicles, milk-derived miRNAs are thought to be uptake by the infant. They may be performing epigenetic regulatory functions during the early neonatal period, including the development of immune protection and brain maturation [8,17]. 

While HBM has unique health advantages for infants, the mechanisms by which it regulates the physiology of newborns are incompletely understood. miRNAs were present in a wide range of fluids. Several highly abundant miRNAs in many fluids may be common among multiple fluid types, and some of the miRNAs were enriched in specific fluids [4]. These specific types and levels of miRNAs, compared to those found in other fluids or even in the cell fraction of HBM [50], support the hypothesis that the miRNA packaging process is a selective rather than a random event [8]. The miRNAs found in HBM originate primarily from mammary gland epithelium [7]. HBM has been highlighted as one of the richest sources of miRNAs in the human body [50]. In contrast, infant formulae are extremely poor in miRNA compared to HBM, with potential differences in the biological activity of these molecules in formula [2].

The 3rd trimester of pregnancy is associated with rapid brain development, including differentiation and maturation of pre-oligodendrocytes, synapses between thalamocortical afferents and subplate neurons, axonal growth and cortical gyrification [51]. It seems that gestational age at delivery influences the composition of preterm milk, which is known to be different from term milk, particularly microRNA in HBM. For example, for preterm infants, who face an increased risk of viral and bacterial infections [7], an enrichment of miRNAs involved in immunologic regulation has been described in HBM [2,16]. The variation in miRNA content and composition during lactation has also been characterized [50,51]. We found an increase in differentially expressed miRNAs when comparing groups according to delivery time. Moderate/very preterm and term colostrum versus mature milk comparisons were the 2nd (27 miRNAs) and the 3rd (24 miRNAs) with more significant differentially expressed miRNAs. In contrast, analysis between mature milk and colostrum, in the term but also moderate/very preterm delivery showed the lowest (11 miRNAs) and the 3rd (19 miRNAs) lower number of miRNAs with significant differential expression.

The late preterm group has the highest number (36 miRNAs) of differentially expressed miRNAs compared to moderate/very preterm mature milk but also the 2nd lower (15 miRNAs) compared to term mature milk. The difference in the number of miRNAs may also be explained based on the delivery time. As late preterm and term mature milk only differ in 1–2 weeks, changes in miRNA composition are minor, whereas moderate/very preterm group milk samples have more time difference.

Preterm delivery may affect the miRNA composition, resulting in a unique HBM miRNA profile. Breast cells alter their miRNA profile due to progesterone and estrogen exposure [52,53]. Estrogen and progesterone levels increase throughout pregnancy, and mothers of preterm babies have lower levels of both hormones at delivery. Thus, this could have a lasting impact on the miRNAs in preterm HBM, providing potential evolutionary benefits for the neonate. This suggests adaptive functions for growth in preterm infants because it is also involved in the glycolytic pathway, glucose metabolism, and obesity-related genes, possibly indicating alterations in the growth and metabolism of preterm infants [7].

In our study, we observed something similar for those miRNAs that regulate the nervous system. The involvement of milk miRNAs in brain development was also suggested by finding that some of the most significant enrichment pathways in miRNAs targets from preterm milk are the biosynthesis of glycosphingolipids and cell membrane function [7]. Preterm infants are also at high risk of neurodevelopmental impairment, possibly due to altered development of brain connectivity and subsequent neurocognitive impairment [51,54]. For example, miR-12136 is significantly downregulated in moderate/very preterm versus term colostrum. miR-12136 abnormal expression can lead to poor brain development causing intellectual disability in the growing fetus [55]. It was also found to be downregulated in plasma from autism spectrum disorder (ASD) patients [56]. miR-320a and miR-320b also have lower levels in moderate/very preterm than in term colostrum. miR-320b preferentially co-localized with neurons in humans and showed strong enrichment in neuron-related genes [57], consistent with our results. miR-320a is downregulated in frontotemporal dementia and AD patients [58], and both isoforms were down-regulated in SCZ compared with cured patients and controls [59].

In the reverse direction, miR-125b-5p is overexpressed compared to moderate/very preterm and term colostrum. It is implicated in promoting neuronal cell differentiation [60,61,62]. Conversely, miR-125b was also highly expressed in patients with AD [63,64,65,66]. Higher levels in moderate/very preterm than in term colostrum for miR-29a, miR-29b and miR-29c also emerged from the data. The three isoforms were described as downregulated in AD brains [67,68]. Also, miR-29a-3p downregulation happens in term colostrum versus term mature milk.

We found a significantly increased moderate/very preterm versus term mature milk miR-23b-3p, which showed a strong ability to enhance neurite outgrowth and nerve regeneration, with *NRP1* as the target gene [69]. In addition, miR-23b-3p is pointed out as a potential therapeutic mechanism in AD because its upregulation inhibited GSK-3β-mediated tau hyperphosphorylation, Aβ generation and neuronal apoptosis [70].

Moreover, miR-1249-3p is also overexpressed in moderate/preterm mature milk compared to term mature milk. Target gene prediction for miR-1249-3p identified two overrepresented canonical pathways involved in axonal growth and ephrin B signaling pathways [71]. GO analysis showed matches for “Neurogenesis”, “Neuron projection morphogenesis”, “Chemical synaptic transmission”, and “Regulation of synaptic plasticity”. miR-1249-3p is downregulated in patients with amyotrophic lateral sclerosis (ALS) compared to controls, and it regulates a specific set of genes associated with the pathophysiology of ALS [71]. According to the distal axonopathy hypothesis of ALS, alterations in the axonal growth may play a fundamental role in ALS as there is accumulating evidence suggesting that the earliest presymptomatic pathological changes occur distally in axons and at the neuromuscular junction [72].

On the contrary direction, miR-4488 has significantly lower levels in preterm than in term mature milk. This miRNA targets synaptotagmin-7 (Syt7), a synapse-associated molecule, and it upregulated considerably in multiple sclerosis (MS) lesions, where Syt7 is maldistributed (accumulated in neuronal soma and decreased in axonal structures). This results in an impairment of neuronal function in MS [73]. Downregulation in preterm versus term mature milk is the same way for miR-423-5p, which is known that acts repressing neurogenesis [74], and also tissue fraction of major depressive disorder (MDD) subjects in the synaptic fraction is upregulated [75]. On the other hand, miR-451a, which has lower levels in late preterm than in term mature milk, was downregulated in AD cerebrospinal fluid (CSF) [65].

miR-146a-5p is downregulated in moderate/very preterm mature milk compared to term mature milk. GO analysis of its targets showed enrichment for the “Regulation of neurogenesis” and “Regulation of dendritic spine development” categories [76], which is consistent with our data (“Synapse” and “Neurogenesis”). miR-146a-5p is upregulated in AD patients’ brains [76,77,78].

miR-17-5p and miR-20a-5p are upregulated in moderate/very preterm colostrum versus moderate/very preterm mature milk and also downregulated in moderate/very preterm mature milk versus term mature milk. miR-17-5p and mir-20a-5p form part of miR-17-92 cluster (miR-17-3p/5p, miR-18, miR-19a/b, miR-20, miR-92a). miR-17-92 cluster is downregulated due to retinoic acid-induced differentiation, showing that it regulates synaptic plasticity and neuronal differentiation markers [79]. miR-17-5p is also increased in the cortex of patients with schizophrenia (SCZ), and target and pathway analysis provided insight into the potential cellular effects of miR-17-5p, with particular enrichment of miRNA targets in axon guidance and long-term potentiation [80].

miR-17 sequences from mammalian species also belong to a second cluster, which contains miR-106a/b [81,82]. miR-106b-5p is also overexpressed in the moderate/very preterm colostrum group compared to term colostrum. A significant decrease in miR-106b expression was found in sporadic AD patients. miR-106, miR-17-5p and miR-20a-5p are lost in cell culture could favoring Aβ formation, inducing neurodegeneration and causing neuronal death [67]. A tight correlation between these miRNAs and APP was found during brain development and in differentiating neurons [83].

The miR-107 family includes miR-16-5p, miR-103a-3p, miR-107, miR-195-5p among others. They have higher levels in moderate/very preterm colostrum than moderate/very preterm mature milk. miR-16-5p is also downregulated in moderate/very preterm mature milk versus term mature milk. miR-103 overexpression in neuroblastoma cells reduces proliferation and promotes differentiation [83]. miR-103 is downregulated in AD patients [84,85] and increased in moderate/very preterm versus moderate/very preterm mature milk. miR-107 is increased in the SCZ cortex. It is involved in the modulation of neuron differentiation regulatory processes. Potential cellular effects, with particular enrichment of miRNA targets in axon guidance and long-term potentiation [80]. miR-107 is significantly higher in SCZ CSF [86]. miR-107 expression tended to be lower than other miRNAs as AD progressed in brain samples’ temporal cortical grey matter [87]. miR-107 levels decreased significantly even in patients with the earliest stages of AD pathology [88]—downregulation of miR-107 in AD grey matter of the middle temporal cerebral cortex [68]. miR-107 is downregulated in AD patients [85]. miR-16 is downregulated in AD tissues and attenuates β-amyloid-induced neurotoxicity [89], and miR-16-5p is downregulated in CSF of AD patients [65].

miR-141-3p and miR-200c are overexpressed in term colostrum versus term mature milk. Their miRNAs repress neural induction of human embryonic stem cells and are downregulated in neuroectodermal precursors cells [90]. In addition, miR-200c is also overexpressed in moderate/very preterm colostrum versus moderate/very preterm mature milk. The same way as miR-30a-5p. miR-30a-5p increase resulted in the downregulation of BDNF protein, suggesting that it is involved in fine-tuning this neurotrophin in adulthood [91].

Local connectivity in brain regions was related to the degree of prematurity and contributed to the altered global topology of the structural brain network [51]. Mothers who deliver before being at the end of the pregnancy produced colostrum, but especially mature milk, with a strong difference in the miRNA content and a specific impact on neuroregulation and neurodevelopment. Having a bigger expression in moderate/very preterm groups of miRNAs that target synaptic components and neuro-related processes may be due to the necessity of a more tuned control over neurodevelopment. Sensibility to nervous system impairments is bigger in moderate/very preterm newborn babies than those born during their maternal pregnancy because brain connectivity and development are unequal. Thus, control over all of these situations needs to be stricter. HBM feeding weeks after preterm birth is associated with improved structural connectivity of developing networks in infants at term-equivalent age [54].

We propose the hypothesis that preterm birth can generate changes in the expression pattern of the miRNAs present in HBM exosomes in such a way that they favor the expression of proneural miRNAs at earlier moments of lactation. The exact influence of breastfed babies of different stages must be investigated and verified with further results. With our data, we propose that there is a mechanism underlying the divergence of the adaptive transcriptome and that changes in miRNA expression could have been an essential driving force to overcome the difficulties that preterm infants may face, contributing to the maturation of the preterm infant—human brain (Figure 24).

In contrast, miRNAs found in term colostrum were similar to those in term mature milk [7]. This is also consistent with our results, where differential expression had only 11 significant miRNAs, the lowest number. It is also quite balanced in terms of upregulated and downregulated, and more clearly when miRNAs are translated to their targets.

Specific miRNAs have versatile functions in both invertebrates and vertebrates during neural development and brain activities. The flexibility, speed, and reversibility of miRNA function provide precise temporal and spatial gene regulatory capabilities that are crucial for the correct functioning of the brain [92]. During prenatal development, the fetus is particularly vulnerable to the effects of a broad range of environmental exposures, with consequences that can persist into infancy, adolescence, and adulthood [93]. Accumulating evidence strongly suggests that dysfunction of miRNAs contributes to neurological diseases. Pathophysiological states that disrupt neuronal networks can lead to neurodevelopmental disorders such as autism, schizophrenia, or intellectual disability [94]. We found miRNAs associated with KEGG terms “Glioma”, “Brain metastatic tumor” and “Synaptic signaling pathway associated with autism spectrum disorder”. Differentially expressed miRNA targets were highly enriched for gene sets related to autism, schizophrenia, bipolar disorder, and depression but not epilepsy, neurodegenerative diseases, or other adult-onset psychiatric diseases [44]. Distinct variations in hub miRNA expression levels or patterns may initiate and/or progress various adult-onset nerve-related diseases [14]. Together with their gene regulation properties, the *CLU* gene and also *CALR* and *KLK6* but less intensity, codify chaperone proteins implicated in Aβ clearance and α-Syn degradation [95,96,97,98,99,100], it implicates miRNAs to be the key regulators in the complex genetic network of the CNS [13].

miRNAs are associated with particular organs, suggesting their participation in conserved developmental or physiological pathways [11]. In the CNS, miRNAs that functionally interact with their target genes constitute a flexible, robust, and buffered regulatory network, exerting diverse roles in regulating brain development [14,44]. Differential expression of miRNAs during neurogenesis has suggested a primary role for miRNA in regulating gene expression during differentiation and development. Although human genomes are known to encode several hundred miRNAs, a significantly smaller subset is either brain-enriched or brain-specific [63]. Data strongly suggest a conserved role for these miRNAs in the specification and/or progression of neuronal fate in mammals, so they could collaborate with or fine-tune the regulation mediated by neurogenic proteins that specify neuronal identity [11]. Many miRNAs detected as differentially expressed between milk groups target genes that codify proteins like *NRP1*, ApoE, *RHOA* or *DPYSL2*, implicated in axonal and neuronal growth [47,48,101,102,103,104,105,106]. In addition, other targets like *CLU* or *KLK6* have an essential role in avoiding brain illness progression [97,107,108,109,110].

One of the possible limitations of the present investigation may be its sample size. We collected 36 samples, and groups ranged from 10 to 4, even though previous studies have utilized a similar number of samples [16,18,25,49,111]. However, it is true that we also found others with a higher number of studied samples [3,7,17]. For powered analysis, more samples may be required, especially for moderate/very preterm colostrum.

The existence of miRNAs in exosomes and their potential function as extracellular regulators have opened up new possibilities for using microRNAs as biomarkers in health and disease and therapeutic modelling. Future perspectives could consist of that miRNA-sequencing should also be performed in the whole milk sample to analyse if it exists differences with exosome cargo. In addition, a longitudinal study would be interesting for preterm mothers to compare how miRNA cargo evolves and if it eventually reaches patterns similar to the term group.

We could extract and sequence the miRNA fraction from HBM exosomes and find differentially expressed miRNAs enriched in several processes, pathways and targets responsible for proper nervous system development after newborn birth.

## 5. Conclusions

microRNAs play beneficial functions in humans and are actively involved in many normal developmental and physiological processes. Together with previous studies, our data demonstrate that miRNAs are abundant in HBM and likely play significant roles in neurodevelopment and normal function. 

After exosome extraction and RNA sequencing, 132 different miRNAs were identified across all samples. Sixty-nine miRNAs had significant differential expression after paired group comparison. We observed differences according to the delivery time and with less clarity according to the milk type.

Based on robust GO and KEGG mapping, 66 miRNAs had at least one match in terms of the pathway (e.g., dopaminergic/glutamatergic synapse, neurotransmitter secretion), biological process (e.g., neuron projection morphogenesis, synaptic vesicle transport,) or cellular component (e.g., presynaptic, and postsynaptic membrane, axon) related with the nervous system. Our data support the notion that these maternally secreted miRNAs play a regulatory role in the infant and account in part for the health benefits of breast milk.

We tentatively propose that these exosomal miRNAs are transferable genetic material from mother to infant and are essential for developing the nervous system in infants. Future studies must demonstrate how individual bioactive components within exosomes exert biological function on the nervous system.

## Figures and Tables

**Figure 1 nutrients-15-03284-f001:**
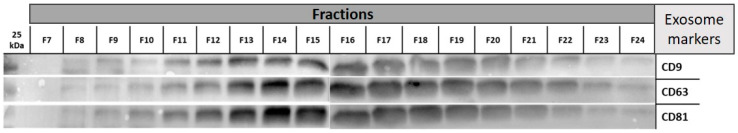
Western blot for exosome-specific surface markers CD9, CD63 and CD81 across exosome-containing fractions extracted with EX04 Exo-spin^TM^ midi column.

**Figure 2 nutrients-15-03284-f002:**
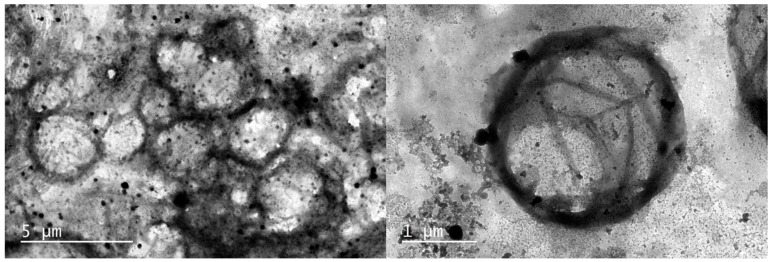
TEM microscopy images of isolated exosomes stained with Phosphotungstic Acid.

**Figure 3 nutrients-15-03284-f003:**
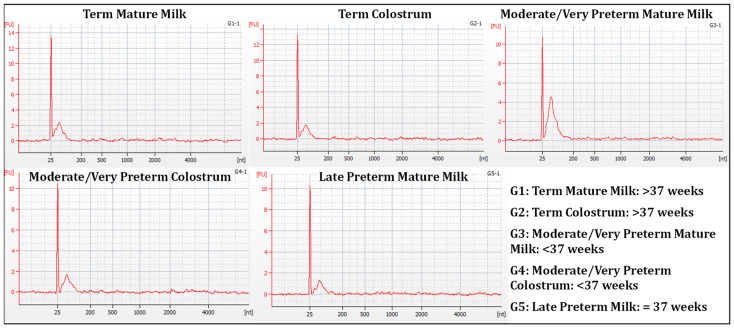
Electropherogram for samples G1 (Term Mature Milk), G2 (Term Colostrum), G3 (Moderate/very Preterm Mature Milk), G4 (Moderate/very Preterm Colostrum) and G5 (Late Preterm Milk) Eukaryote Total RNA Nano Assay.

**Figure 4 nutrients-15-03284-f004:**
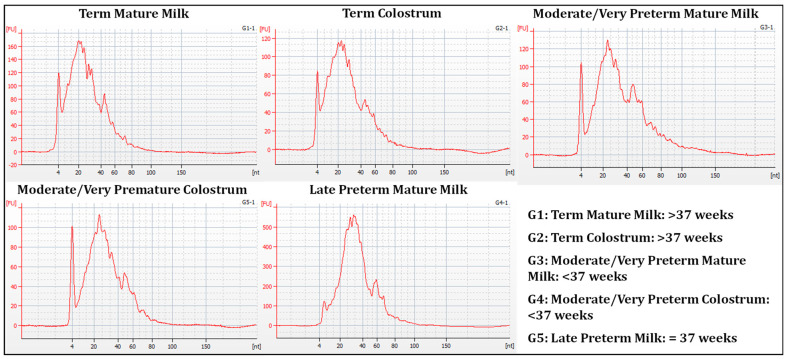
Electropherogram for samples G1 (Term Mature Milk), G2 (Term Colostrum), G3 (Moderate/very Preterm Mature Milk), G4 (Moderate/very Preterm Colostrum) and G5 (Late Preterm Milk) Small RNA Assay.

**Figure 5 nutrients-15-03284-f005:**
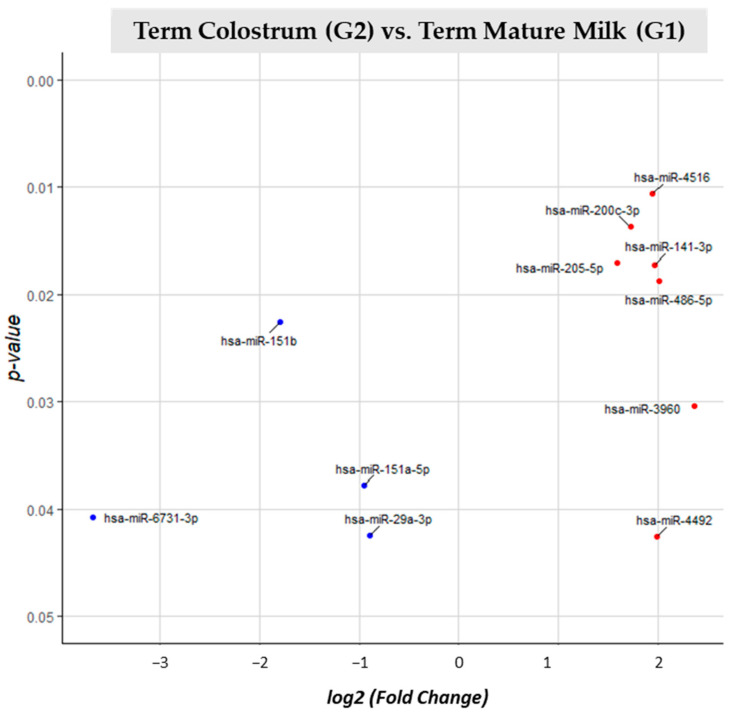
Volcano plot depicting the significant differentially expressed miRNAs from Term Colostrum (G2) vs. Term Mature Milk (G1). miRNAs with red dots are upregulated, miRNAs with blue dots are downregulated in G2 vs. G1.

**Figure 6 nutrients-15-03284-f006:**
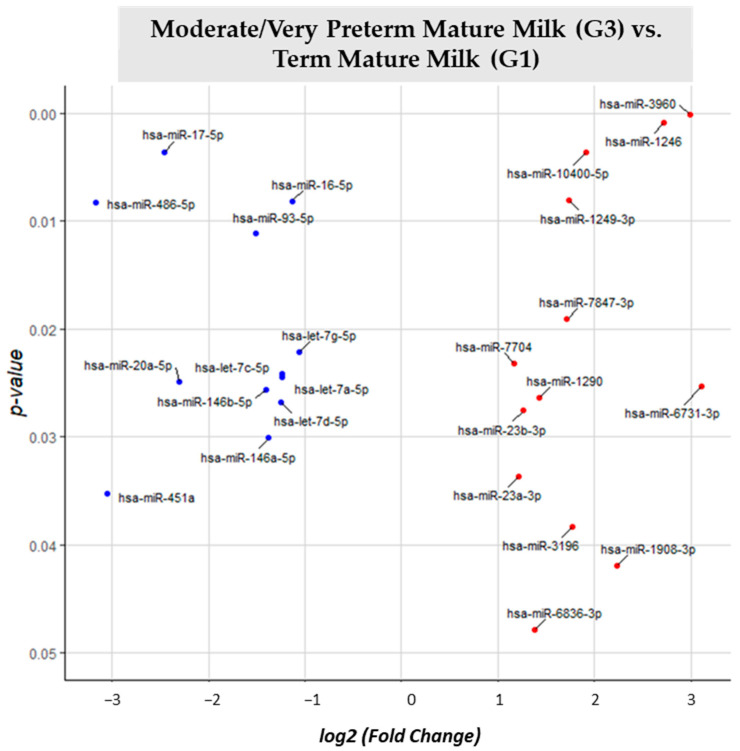
Volcano plot depicting the significant differentially expressed miRNAs from Moderate/Very Preterm Mature Milk (G3) vs. Term Mature Milk (G1). miRNAs with red dots are upregulated, miRNAs with blue dots are downregulated in G3 vs. G1.

**Figure 7 nutrients-15-03284-f007:**
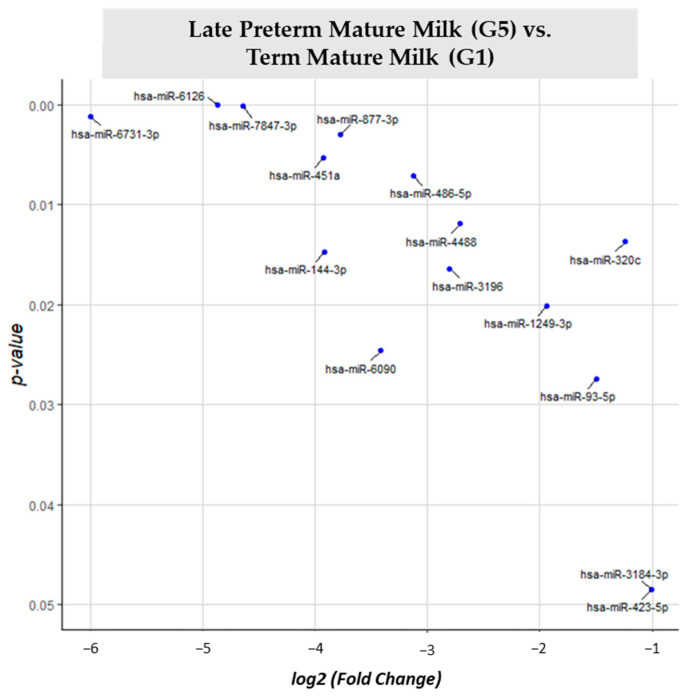
Volcano plot depicting the significant differentially expressed miRNAs from Late Preterm Mature Milk (G5) vs. Term Mature Milk (G1). miRNAs with blue dots are downregulated in G5 vs. G1.

**Figure 8 nutrients-15-03284-f008:**
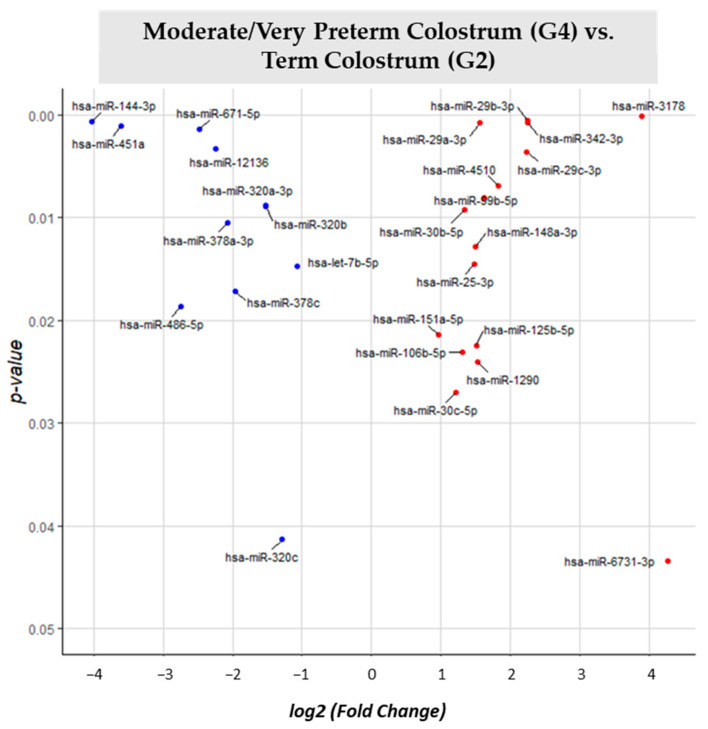
Volcano plot depicting the significant differentially expressed miRNAs from Moderate/Very Preterm Colostrum (G4) vs. Term Colostrum (G2). miRNAs with red dots are upregulated, miRNAs with blue dots are downregulated in G4 vs. G2.

**Figure 9 nutrients-15-03284-f009:**
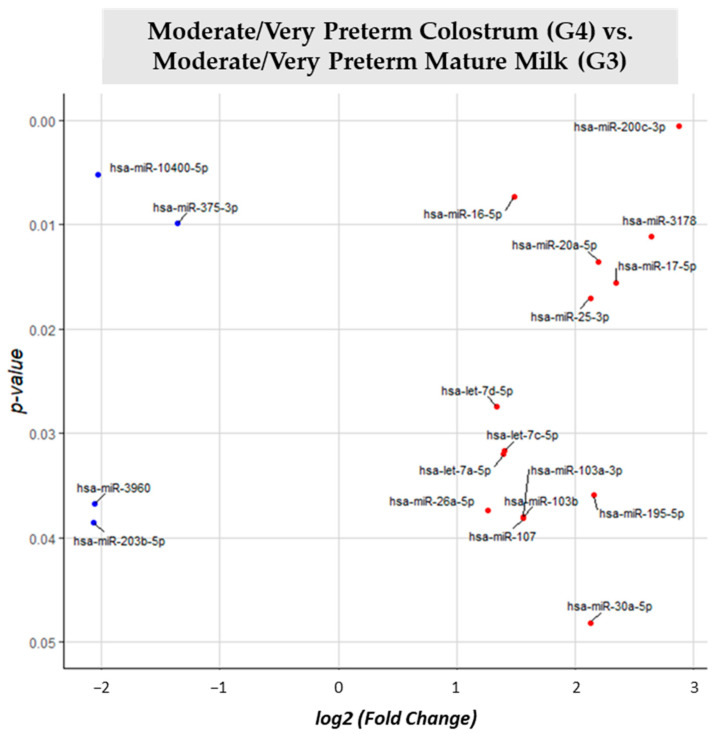
Volcano plot depicting the significant differentially expressed miRNAs from Moderate/Very Preterm Colostrum (G4) vs. Moderate/Very Preterm Mature Milk (G3). miRNAs with red dots are upregulated, miRNAs with blue dots are downregulated in G4 vs. G3.

**Figure 10 nutrients-15-03284-f010:**
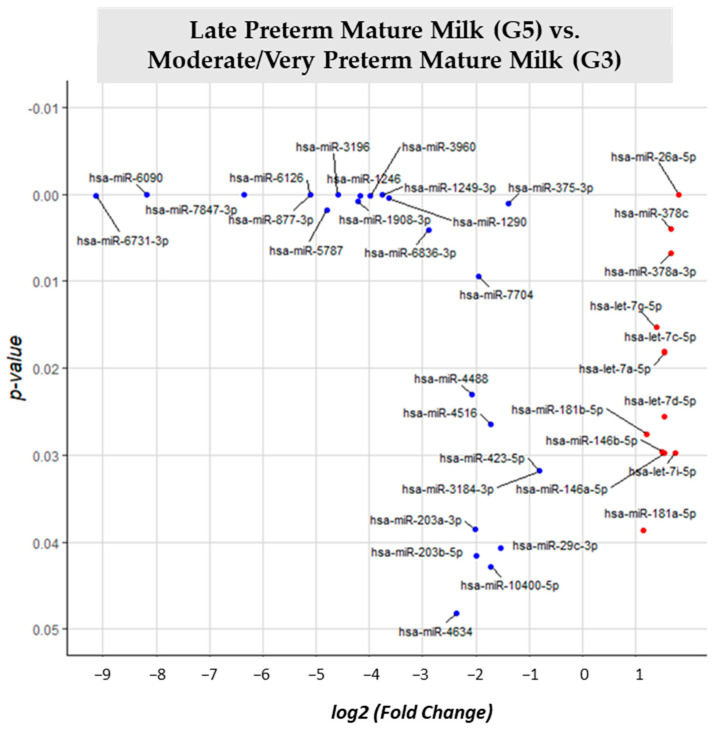
Volcano plot depicting the significant differentially expressed miRNAs from Late Preterm Mature Milk (G5) vs. Moderate/Very Preterm Mature Milk (G3). miRNAs with red dots are upregulated, miRNAs with blue dots are downregulated in G5 vs. G3.

**Figure 11 nutrients-15-03284-f011:**
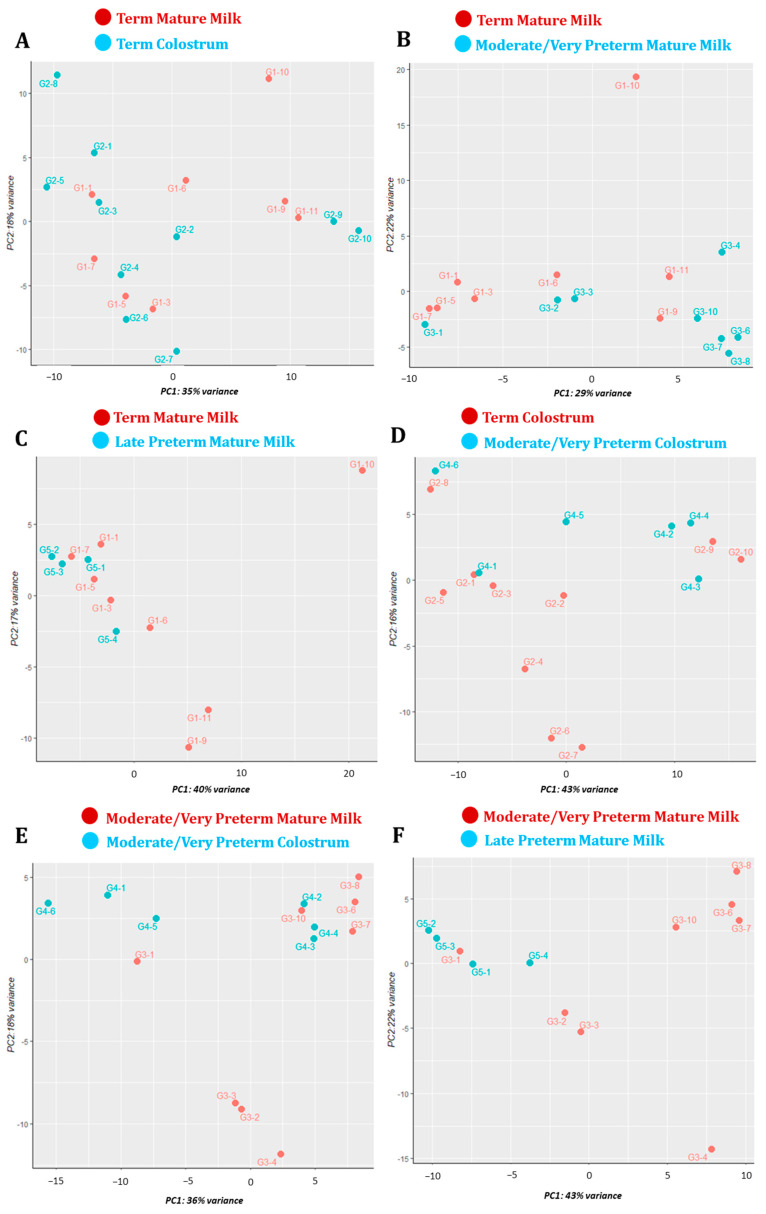
PCA plot for samples in (**A**) Term Mature Milk (G1) and Colostrum (G2) groups. (**B**) Term (G1) and Moderate/Very Preterm (G3) Mature Milk groups. (**C**) Term (G1) and Late Preterm (G5) Mature Milk groups. (**D**) Term (G2) and Moderate/Very Preterm (G4) Colostrum groups. (**E**) Moderate/Very Preterm Mature Milk (G3) and Colostrum (G4) groups. (**F**) Moderate/Very Preterm (G3) and Late Preterm (G5) Mature Milk groups.

**Figure 12 nutrients-15-03284-f012:**
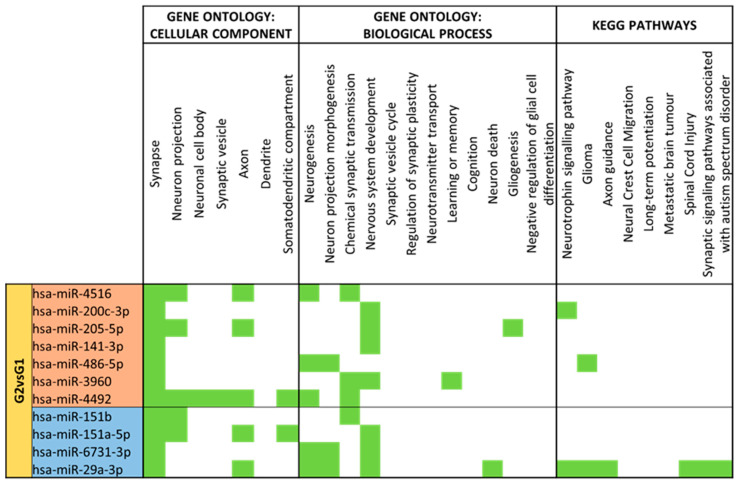
miRNA target analysis in terms of Cellular Component, Biological Process and KEGG Pathways for significant differentially expressed miRNAs for Term Colostrum (G2) vs. Mature Milk (G1). miRNAs with red backgrounds are upregulated, whereas miRNAs with blue backgrounds are downregulated in G2 compared to G1. In green it is represented when the miRNA localizes within the pathways in terms of Cellular Component, Biological Process and KEGG.

**Figure 13 nutrients-15-03284-f013:**
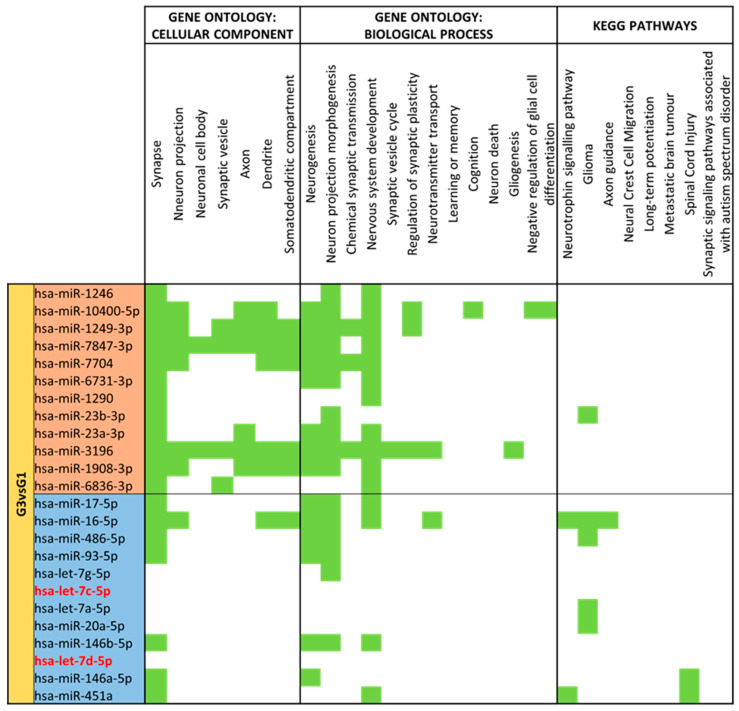
miRNA target analysis in terms of Cellular Component, Biological Process and KEGG Pathways for significant differentially expressed miRNAs for Moderate/Very Preterm (G3) vs. Term (G1) Mature Milk. miRNAs with a red background are upregulated, whereas miRNAs with a blue background are downregulated in G3 compared to G1. In green it is represented when the miRNA localizes within the pathways in terms of Cellular Component, Biological Process and KEGG.

**Figure 14 nutrients-15-03284-f014:**
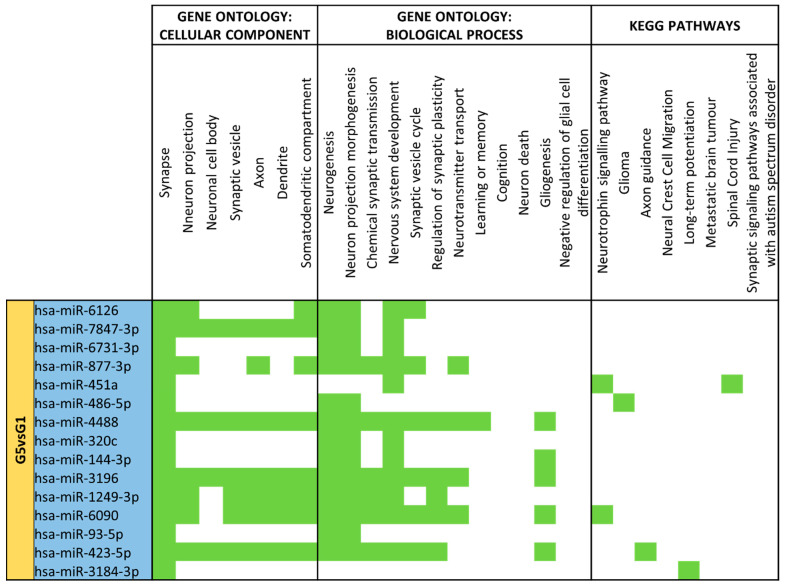
miRNA target analysis in terms of Cellular Component, Biological Process and KEGG Pathways for significant differentially expressed miRNAs for Late Preterm (G5) vs. Term (G1) Mature Milk. miRNAs with a blue background are downregulated in G5 compared to G1. In green it is represented when the miRNA localizes within the pathways in terms of Cellular Component, Biological Process and KEGG.

**Figure 15 nutrients-15-03284-f015:**
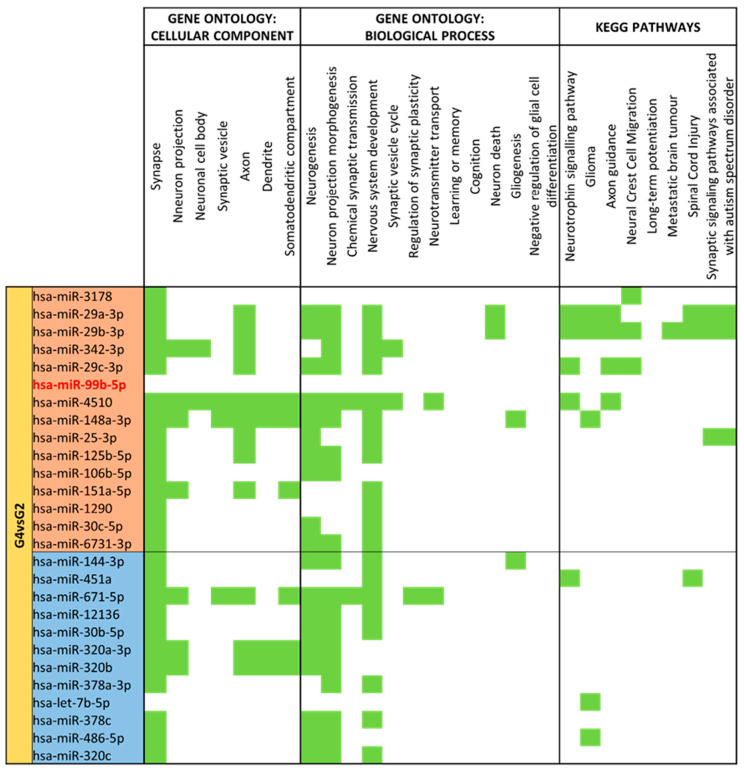
miRNA target analysis in terms of Cellular Component, Biological Process and KEGG Pathways for significant differentially expressed miRNAs for Moderate/Very Preterm (G4) vs. Term (G2) Colostrum. miRNAs with red backgrounds are upregulated, whereas miRNAs with blue backgrounds are downregulated in G4 compared to G2. In green it is represented when the miRNA localizes within the pathways in terms of Cellular Component, Biological Process and KEGG.

**Figure 16 nutrients-15-03284-f016:**
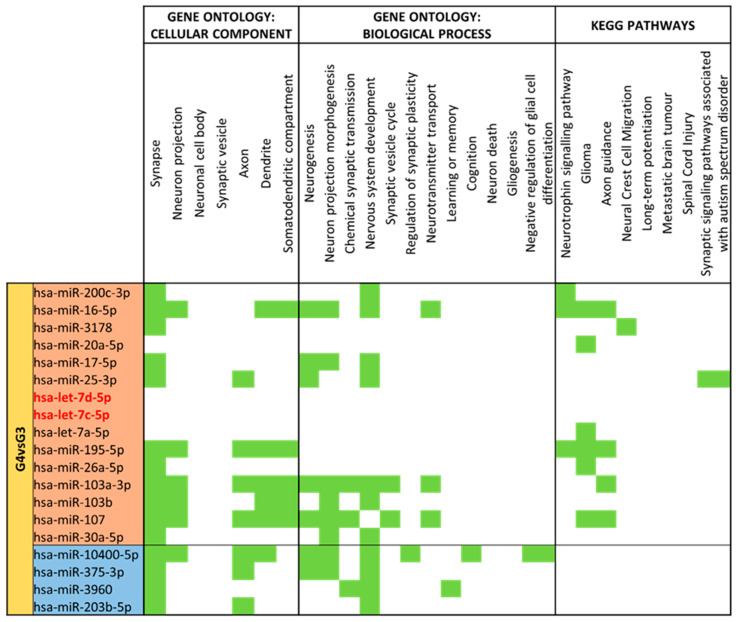
miRNA target analysis in terms of Cellular Component, Biological Process and KEGG Pathways for significant differentially expressed miRNAs for Moderate/Very Preterm Colostrum (G4) vs. Mature Milk (G3). miRNAs with a red background are upregulated, whereas miRNAs with a blue background are downregulated in G4 compared to G3. In green it is represented when the miRNA localizes within the pathways in terms of Cellular Component, Biological Process and KEGG.

**Figure 17 nutrients-15-03284-f017:**
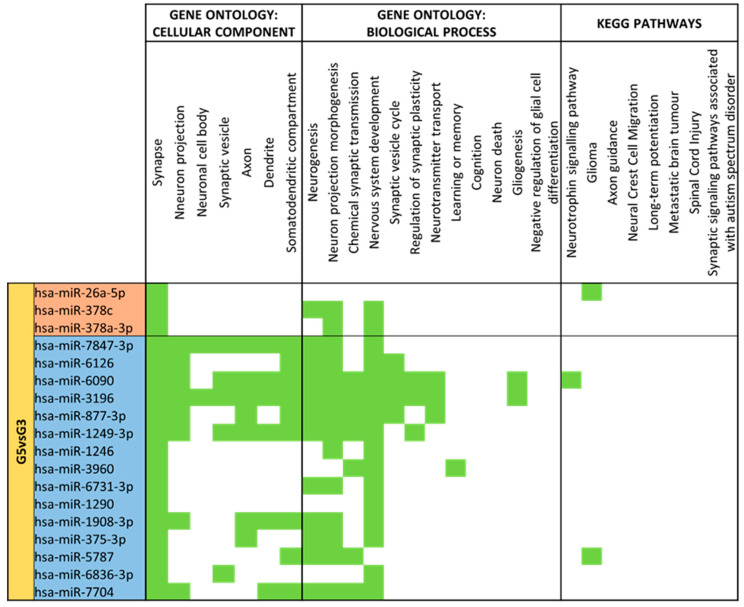
miRNA target analysis in terms of Cellular Component, Biological Process and KEGG Pathways for significant differentially expressed miRNAs for Late Preterm (G5) vs. Moderate/Very Preterm (G3) Mature Milk. miRNAs with a red background are upregulated, whereas miRNAs with a blue background are downregulated in G5 compared to G3. In green it is represented when the miRNA localizes within the pathways in terms of Cellular Component, Biological Process and KEGG.

**Figure 18 nutrients-15-03284-f018:**
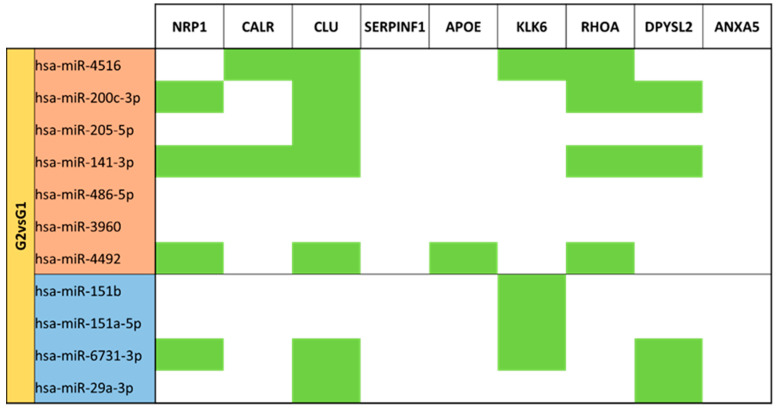
miRNA target analysis over codifying genes of 9 neuro-related proteins for the significant differentially expressed miRNAs in Term Colostrum (G2) vs. Mature Milk (G1). miRNAs in the red background are upregulated, and miRNAs in the blue background are downregulated. Green cells mean the relationship between genes and miRNA.

**Figure 19 nutrients-15-03284-f019:**
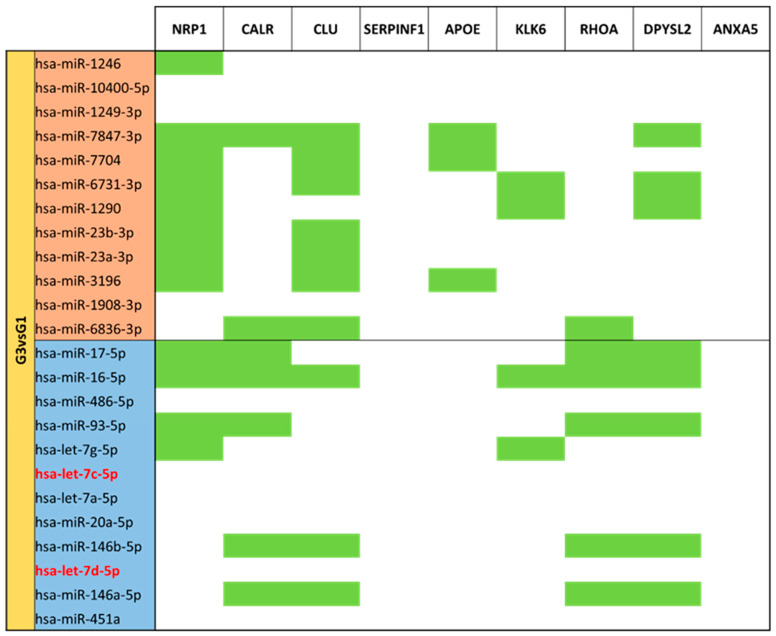
miRNA target analysis over codifying genes of 9 neuro-related proteins for the significant differentially expressed miRNAs in Moderate/Very Preterm (G3) vs. Term (G1) Mature Milk. miRNAs with red background are upregulated, miRNAs with blue background are downregulated. Green cells mean the relationship between genes and miRNA. Green cells mean the relationship between genes and miRNA.

**Figure 20 nutrients-15-03284-f020:**
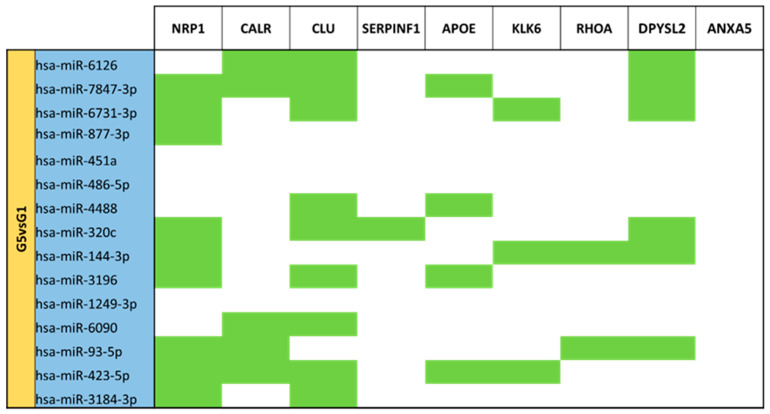
miRNA target analysis over codifying genes of 9 neuro-related proteins for the significant differentially expressed miRNAs in Late Preterm (G5) vs. Term (G1) Mature Milk. miRNAs with a blue background are downregulated. Green cells mean the relationship between genes and miRNA.

**Figure 21 nutrients-15-03284-f021:**
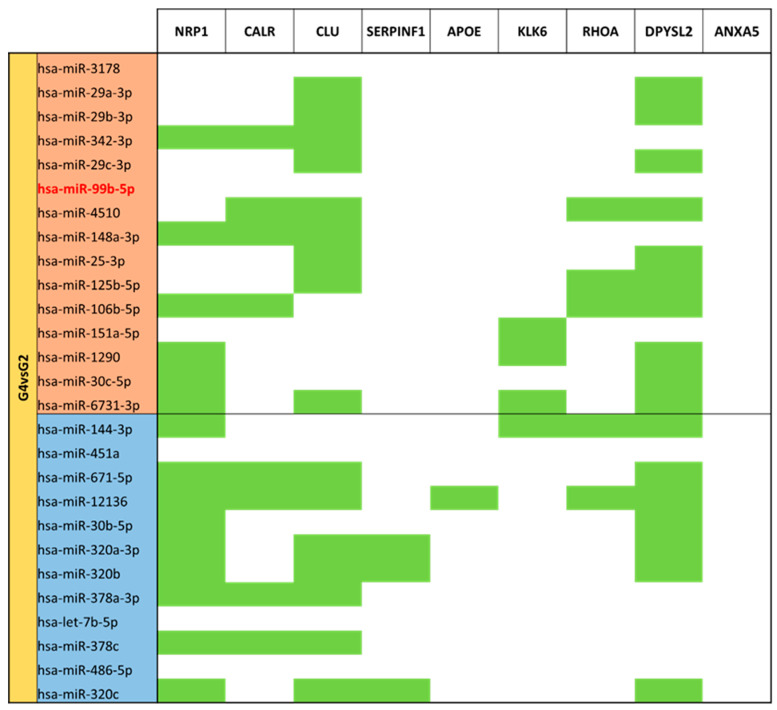
miRNA target analysis over codifying genes of 9 neuro-related proteins for the significant differentially expressed miRNAs in Moderate/Very Preterm (G4) vs. Term (G2) Colostrum. miRNAs with red background are upregulated, miRNAs with blue background are downregulated. Green cells mean the relationship between genes and miRNA.

**Figure 22 nutrients-15-03284-f022:**
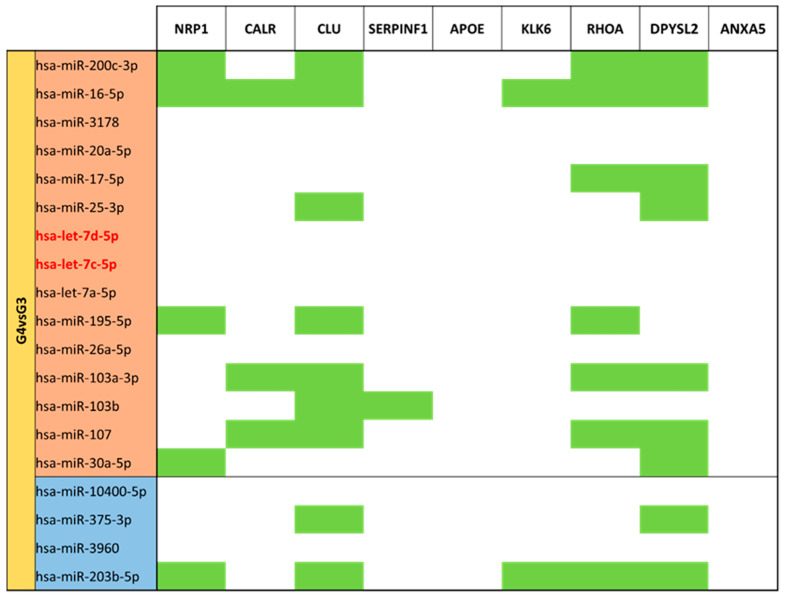
miRNA target analysis over codifying genes of 9 neuro-related proteins for the significant differentially expressed miRNAs in Moderate/Very Preterm Colostrum (G4) vs. Mature Milk (G3). miRNAs with red background are upregulated, miRNAs with blue background are downregulated. Green cells mean the relationship between genes and miRNA.

**Figure 23 nutrients-15-03284-f023:**
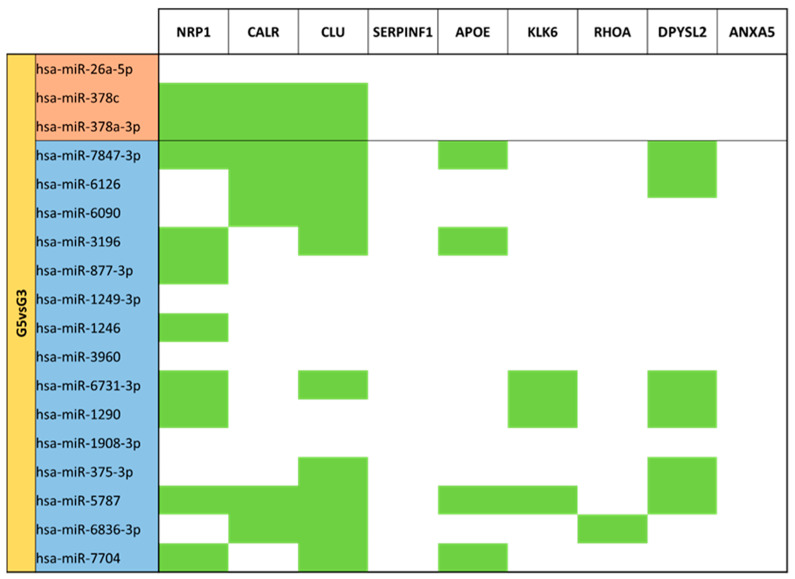
miRNA target analysis over codifying genes of 9 neuro-related proteins for the significant differentially expressed miRNAs in Late Preterm (G5) vs. Moderate/Very Preterm (G3) Mature Milk. miRNAs with red background are upregulated, miRNAs with blue background are downregulated. Green cells mean the relationship between genes and miRNA.

**Figure 24 nutrients-15-03284-f024:**
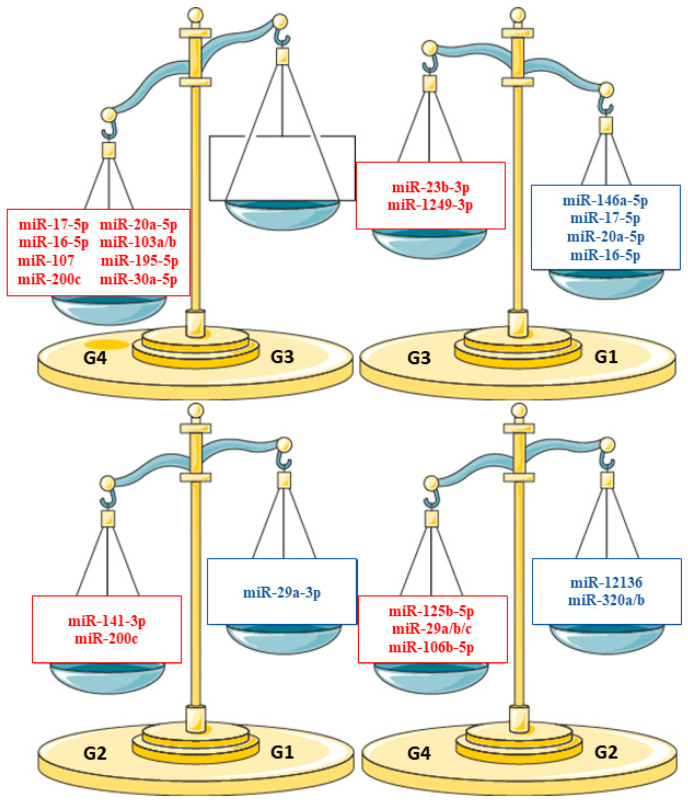
Balances model about miRNA expression changes.

**Table 1 nutrients-15-03284-t001:** Sample ID, Group, RNA concentration, Ratio 260/280 and 260/260, RIN and % of miRNA fraction.

Sample ID	Multiplexing Group	[RNA] (ng/µL)	R 260/280	R 260/230	RIN	miRNA Fraction
G1-1	Term Mature Milk	39.97	1.725	0.303	1.8	87%
G1-3	Term Mature Milk	20.69	1.631	0.455	1.1	90%
G1-5	Term Mature Milk	39.87	1.929	0.793	2.6	76%
G1-6	Term Mature Milk	16.95	1.606	0.105	1	85%
G1-7	Term Mature Milk	24.96	1.545	0.576	1.4	85%
G1-9	Term Mature Milk	10.601	1.75	0.412	1	93%
G1-10	Term Mature Milk	13.71	1.539	0.161	1	91%
G1-11	Term Mature Milk	19.62	1.484	1.043	2	93%
G2-1	Term Colostrum	19.72	1.806	0.349	1.6	86%
G2-2	Term Colostrum	14.63	1.62	0.289	5.4	83%
G2-3	Term Colostrum	22.32	1.476	0.231	1	87%
G2-4	Term Colostrum	20.53	1.639	0.616	1	80%
G2-5	Term Colostrum	16.31	1.963	0.629	2.2	80%
G2-6	Term Colostrum	28.32	1.924	0.797	2.4	89%
G2-7	Term Colostrum	31.35	1.85	0.737	2.6	89%
G2-8	Term Colostrum	10.812	1.694	0.449	1	43%
G2-9	Term Colostrum	10.894	1.478	0.805	1	84%
G2-10	Term Colostrum	12.55	1.468	0.528	1	82%
G3-1	Moderate/Very Preterm Mature Milk	16.31	1.963	0.629	2.5	80%
G3-2	Moderate/Very Preterm Mature Milk	12.55	1.489	0.337	1	56%
G3-3	Moderate/Very Preterm Mature Milk	10.821	1.687	0.045	1	57%
G3-4	Moderate/Very Preterm Mature Milk	10.739	1.904	0.36	1	60%
G3-6	Moderate/Very Preterm Mature Milk	10.853	1.876	0.364	1	90%
G3-7	Moderate/Very Preterm Mature Milk	13.23	1.433	0.703	1	89%
G3-8	Moderate/Very Preterm Mature Milk	10.058	1.546	0.386	1	90%
G3-10	Moderate/Very Preterm Mature Milk	10.023	1.662	0.284	1	85%
G4-1	Moderate/Very Preterm Colostrum	16	1.818	0.426	1.7	84%
G4-2	Moderate/Very Preterm Colostrum	11.55	1.53	0.526	1	87%
G4-3	Moderate/Very Preterm Colostrum	14.43	1.634	0.401	1	88%
G4-4	Moderate/Very Preterm Colostrum	10.512	1.726	0.426	1	91%
G4-5	Moderate/Very Preterm Colostrum	10.91	1.786	0.264	1	83%
G4-6	Moderate/Very Preterm Colostrum	11	1.571	0.158	1	54%
G5-1	Late Preterm Mature Milk	26.61	1.641	0.636	1.3	86%
G5-2	Late Preterm Mature Milk	27.02	1.9	0.606	2.6	68%
G5-3	Late Preterm Mature Milk	19.71	1.806	0.387	1.2	53%
G5-4	Late Preterm Mature Milk	18.42	1.915	0.426	1.3	72%

**Table 2 nutrients-15-03284-t002:** DESeq2 Software Comparisons, Reference Group for each and number of significant miRNAs. T: Term; MVPT: Moderate/very Preterm; LPT: Late Preterm.

Comparison	Reference Group	Significant Mirnas
T Colostrum (G2) vs. T Mature Milk (G1)	G1	11
MVPT Mature Milk (G3) vs. T Mature Milk (G1)	G1	24
LPT Mature Milk (G5) vs. T Mature Milk (G1)	G1	15
MVPT Colostrum (G4) vs. T Colostrum (G2)	G2	27
MVPT Colostrum (G4) vs. MVPT Mature Milk (G3)	G3	19
LPT Mature Milk (G5) vs. MVPT Mature Milk (G3)	G3	36

## Data Availability

The data presented in this study are available upon reasonable request from the corresponding author. The data are not publicly available due to personal data protection.

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
