# Peer review of "Human Breast Milk microRNAs, Potential Players in the Regulation of Nervous System"

_nutrients, 2023, doi:10.3390/nu15143284_

Round 1
Reviewer 1 Report
Different stages of human breast milk microRNAs were analyzed in the present manuscript. The results provided valuable information for further study. Several suggestions are as below.
1. According to miRNA target analysis (Fig 12-17), not only neuro-related pathways were significantly different. The reason why the further investigation focused on neuro-related proteins needs to be explained.
2. Fig 24 pointed out the modulatory capacity between the expression changes of miRNAs clearly. However, the exact influence for infants fed with breast mike of different stages needs further verified. Several statements in Discussion seemed not very appropriate. The authors maybe could consider to be more conservative to reserve opportunity for future study.
Overall, the present manuscript reported a superior discovery; and is very valuable for future study.
Reviewer 2 Report
It is not clear throughout the study which fractions of exosome isolates where used for RNA isolation and secondly, if the RNA was isolated after these fractions were united.
English language quality is quite good. However, some modifications are necessary to enhance precision and clarity.
Reviewer 3 Report
Comments
1. Please define all abbreviations in the text when used for the first time. For example, in line 29 “miRNAs”.
2. Although miRNAs identified in this sudy likely play significant roles in the neurodevelopment and normal function. This conclusion is not based on experimental data in this research. Further experiments in vitro and even in vivo are necessary to verify this conclusion. The following research directions are my suggestions. a. Human breast milk exosome neuronal cells were co-cultured with exosomes, detect the biological changes of neuronal cells; b. Up-regulate or down-egulate the expression of target miRNAs in exosome, detect the biological changes of neuronal cells.
3. Figure 11 and Figure are not clear, please provide the clear figures.
4. “microRNAs” in title is the microRNAs from Human Breast milk or Human Breast milk exosome. Please confirm that.
Round 2
Reviewer 3 Report
This manuscript can be accepted as present form.